# Controllable Drug Release Behavior of Polylactic Acid (PLA) Surgical Suture Coating with Ciprofloxacin (CPFX)—Polycaprolactone (PCL)/Polyglycolide (PGA)

**DOI:** 10.3390/polym12020288

**Published:** 2020-02-01

**Authors:** Shuqiang Liu, Juanjuan Yu, Huimin Li, Kaiwen Wang, Gaihong Wu, Bowen Wang, Mingfang Liu, Yao Zhang, Peng Wang, Jie Zhang, Jie Wu, Yifan Jing, Fu Li, Man Zhang

**Affiliations:** 1College of Textile Engineering, Taiyuan University of Technology, Taiyuan 030024, China; yuxiaoyuy@126.com (J.Y.); wo1553557384@163.com (H.L.); wkw1011@163.com (K.W.); wangbowen316@163.com (B.W.); lmf118119@163.com (M.L.); zy32164872@126.com (Y.Z.); tyut19834520797@163.com (P.W.); Zj13283543186@163.com (J.Z.); 18234506489@163.com (J.W.); mary1123@126.com (Y.J.); lifu@tyut.edu.cn (F.L.); zhangman@tyut.edu.cn (M.Z.); 2Biomedical Textile Laboratory, Taiyuan University of Technology, Jinzhong 030600, China

**Keywords:** polylactic acid suture, ciprofloxacin, polyglycolide, polycaprolactone, drug release

## Abstract

Polylactic acid (PLA) surgical suture can be absorbed by human body. In order to avoid surgical site infections (SSIs), the drug is usually loaded on the PLA suture, and then the drug can release directly to the wound. Because the different types of wounds heal at different times, it is needed to control the drug release rate of PLA suture to consistent to the wound healing time. Two biopolymers, polyglycolide (PGA) and polycaprolactone (PCL), were selected as the carrier of ciprofloxacin (CPFX) drug, and then the CPFX-PCL/PGA was coated on the PLA suture. The degradation rate of drug-carrier can be controlled by adjusting the proportion of PCL/PGA, which can regulate the rate of CPFX drug release from PLA suture. The results show that the surface of PLA suture, coating with PCL/PGA, was very rough, which led to increased stitching resistance when we were suturing the wound. These materials, such as the PLA suture, the PCL/PGA carriers and the CPFX drug, were just physically mixed rather than chemically reacted, which was very useful for ensuring the original efficacy of CPFX drug. With the increasing of PCL in the carriers, both the breaking strength and elongation of these un-degraded sutures increased. During degradation, the breaking strength of all sutures gradually decreased, and the more PCL in the coating materials, the longer effective strength-time for the suture. With the increasing of PCL in the drug-carrier, the rate of drug releasing became lower. The drug release mechanism of CPFX-PCL/PGA was a synergistic effect of drug diffusion and PCL/PGA carrier dissolution.

## 1. Introduction

Polylactic acid (PLA) polymer has some good properties, such as biocompatibility, biodegradability, non-toxicity and high strength, so it is usually used as the biomedical and pharmaceutical material [1,2,3]. PLA has a good compatibility with human tissues, and can be gradually degraded into safe products of CO_2_ and H_2_O in vivo. Moreover, the intermediate product of PLA is lactic acid (LA), which is the normal sugar metabolite and does not accumulate in human’s vital organs. Therefore, the filaments, which are made of PLA, can be used as the suture which can be absorbed by human body [4,5,6].

The PLA surgical suture has been widely used in the medical field, but there is a problem of postoperative infection. The reduction of surgical site infections (SSIs) promises to be an area of intense interest and activity in the foreseeable future [7,8,9]. In order to avoid SSIs, it is a common method to take a large number of drugs by injection or orally. However, those drugs, which are applied throughout the body, have some great side-effects on human’s tissues and organs. Therefore, it is a straightforward and effective method to load the drugs on the PLA suture, which can release the drug directly to the wound [10,11,12].

In the medical field, there is a requirement for the drug release of drug-loaded surgical suture. Because the different types of wounds would heal at different times, the release time of drug from surgical suture should be consistent or close to the wound healing time. If the drug release time of suture is much longer than the wound healing time, this will be equivalent to overdose, which will cause some toxic and side effects on the tissues. If the drug release time of suture is much shorter than the wound healing time, the drugs will not play a role in the treatment of wound. Therefore, it is need to control the drug release time of surgical suture, so as to be consistent with the different healing time of wounds.

Currently, some experts have used different methods to load drug onto the surgical suture. Champeau et al. [13] studied the relationship between three implant polymer sutures and two anti-inflammatory drugs under supercritical CO_2_. They used a simple dipping method to load the drug onto the suture, and focused on the effect of experimental conditions on drug-loading, rather than how to control the drug release. Lee et al. [14] demonstrated that the drug of dexamethasone can be attached to the lactide suture in the form of microspheres. This suture has an anti-inflammatory effect, but does not control the release of drug. Weldon et al. [15] reported a method for placing the drug of aspirin into PLA/RSF nano-fibers by electro-spinning, but this method is still unable to control the drug release. In conclusion, these drug-loading methods, applied in the previous research results, are not yet able to control the rate and cycle of drug release from suture.

In order to control the rate and cycle of drug release from PLA suture, we applied a new method of drug delivery in this paper, as shown in Figure 1. Two biopolymers, polyglycolide (PGA) and polycaprolactone (PCL), are selected as the carrier of drug in this paper. The biopolymers of PGA and PCL are mixed with the drug to make a mixed liquid. Then the mixed liquid is coated onto the surface of PLA suture by dip rolling process. Finally, the PLA suture coating with PGA, PCL and the drug are prepared.

It should be noted that the biopolymers of PGA and PCL, which are the carrier of drug and coated on the suture, have different degradation rate. The degradation rate of PGA is higher than that of PCL [16,17,18]. We can control the degradation rate of drug-carrier by adjusting the proportion of PGA and PCL. As the drug-carrier of PGA/PCL degrades, the internal drugs are released. Therefore, the proportion of PGA and PCL is an important factor to regulate the rate and cycle of drug release from suture. In addition, the drug coated on the suture is Ciprofloxacin (CPFX), which has a strong antibacterial effect against gram-positive bacteria by inhibiting the gyrase of one of the bacterial DNA synthetase enzymes [19,20,21].

In this paper, the tensile properties of sutures with different ratio of PCL/PGA were measured and the chemical structure of coating was characterized. In the process of degradation and drug release, the surface morphology of the sutures were observed, the time of the suture strength failure, the release rate and cycle of the drug were recorded, and the release kinetics model was established. Besides the mechanism of drug release by sutures were analyzed. We conducted these investigations to provide a theoretical basis for the development of drug-loaded PLA surgical sutures with excellent controlled-release properties, small side effects, and safe and reliable clinical application.

## 2. Materials and Methods

### 2.1. Materials

The original PLA surgical suture, with 33.72 tex, breaking strength: 32.51 cN/tex, elongation at break: 28.48%, was produced by Zhejiang Gaoxin Company (Jiaxing, China).

The drug carrier was made from PGA ((C_4_H_4_O_4_)_n_, *M*_w_ = 11,602 g/mol) and PCL ((C_6_H_10_O_2_)_n_, *M*_w_ = 50,000 g/mol), which were supplied by Natureworks Company (Blair, NE, USA).

The drug used in this paper is CPFX (C_17_H_18_FN_3_O_3_, purity 98%), which was purchased from North China biotechnology Co., Ltd., (Zhengzhou, China). CPFX is a new quinolone broad-spectrum antimicrobial agent, which has bactericidal effect on gram-positive and negative bacteria including pseudomonas aeruginosa, intestinal bacteria and staphylococcus aureus. In this paper, CPFX was loaded onto the suture to prevent wound inflammation.

Some chemical reagents were applied in this experiment frequently. For instance, ethyl acetate (CH_3_COOC_2_H_5_, purity 99.8%), absolute ethyl alcohol (C_2_H_6_O, purity 99.5%), medical alcohol (C_2_H_6_O, purity 75%), Tween-80 (C_2_H_44_O_6_(C_2_H_4_O)n, purity 99%) and Propanetriol (CH_2_OHCHOHCH_2_OH, purity 99%) were provided by Yongda Chemical Reagent Company (Tianjin, China). Ferrous tartrate solution (C_4_H_4_FeO_6_) was obtained from Zhen Jie Quality Inspection Technology Service Company (Guangzhou, China). Phosphate buffer salt (PBS) solution (pH = 7.4) were procured from Sigma Aldrich Company (Shanghai, China).

### 2.2. Preparation of Drug-Loaded Suture

The original PLA surgical sutures should be pretreated firstly. The original PLA sutures were soaked in absolute ethyl alcohol for 2 h to remove the oil on the surface of suture. After that, the sutures were heat setting in vacuum oven at 60 °C for 30 min. Then the sutures were impregnated and disinfected in medical alcohol at room temperature for 10 min. Finally, the sutures were dried in the blast oven at 40 °C for 60 min.

We dissolved 3 g of PCL and PGA in different proportions in 40 mL Ethyl Acetate (purity 99.8%). The carrier solution was prepared by heating and stirring to dissolve PCL and PGA. 1.5 g of Ciprofloxacin (CPFX) and an emulsifier (Propanetriol, Tween-80) were dissolved in 40 mL of distilled water to prepare a mixed water solution of drug (CPFX). The water solution of drug (CPFX) was slowly added to the carrier solution of PCL/PGA. The two solutions were fully emulsified, and a drug-coating finishing liquid of CPFX-PCL/PGA was obtained.

The pretreated PLA surgical suture was treated by the liquid of CPFX-PCL/PGA in the process of dip-padding (Figure 2). After that, the suture was dried in a vacuum oven at 45 °C for 2 h, and then the PLA suture was coated with CPFX-PCL/PGA.

### 2.3. Characterization

The mechanical properties of sutures were tested by electronic single yarn strength tester (YG061FQ, Laizhou Electron Instrument Co., Ltd., Laizhou, China). The clamping length was 250 mm, and the drawing speed was 250 mm/min.

FTIR spectra of sutures were measured by total reflectance infrared spectroscopy and Fourier infrared spectrometer (Tensor27, Bruker Company, Karlsruhe, Germany). The scanning range was 4000–600 cm^−1^, resolution 4 cm^−1^.

The surface morphology of drug-loaded sutures was observed by scanning electron microscopy (JEM2100F, JEOL, Tokyo, Japan) under an acceleration voltage of 10 kV. All the samples’ surfaces were sputtered with gold.

The degradation and drug-release experiments of sutures were performed in a sterile room, where 12 h of UV light per day was used to kill bacteria. The coated sutures were placed in a jar which was containing 50 mL PBS buffer solution. Then the suture was degraded and released drug in vitro by immersing in buffer solution in a constant temperature water bath at (37 ± 0.5) °C. After a certain amount of time, the sutures were taken out and dried, and then the surface morphology, strength and other properties of sutures during degradation were measured.

### 2.4. Drug Release Behavior In Vitro

The absorption intensity of ciprofloxacin (CPFX) was measure by an UV-Vis Spectra Photometer (UV-752, Shanghai Yoke Instrument Company, Shanghai, China). The absorbance of CPFX solution with different concentration was determined at 277 nm wavelength, and 0.1mol/L hydrochloric acid solution was used as blank control. The standard curve can be obtained by using absorbance (y) as vertical coordinate and CPFX concentration (x) as horizontal coordinate, as shown in Figure 3. The linear regression equation of the standard curve obtained in this experiment is y = 1.213x − 0.0007 (*R*^2^ = 0.997).

The drug was released during the degradation of the PLA suture. At the predetermined time, 1 mL of each sample solution was extracted to test the drug content. Meanwhile, an equal amount of fresh buffer was supplemented to each jar [22,23]. The amount of drug release was determined by standard curve of ciprofloxacin and the cumulative release rate was calculated according to Formula (1).
(1)$$Q_{t}=m_{t}/m_{o}\times 100\%$$
where *Q_t_* represented the cumulative release rate of ciprofloxacin at *t* time; *m_t_* represented the content of released ciprofloxacin at *t* time; *m*_0_ represented the content of ciprofloxacin at initial condition.

## 3. Results and Discussion

### 3.1. Surface Morphology of Drug-Loaded PLA Suture

The surface morphology of the original PLA suture and the PLA suture coating with 50/50 PCL/PGA was observed by SEM, as shown in Figure 4.

It can be seen from Figure 4a that the fibers in original PLA suture, which had no coating, were very loose. In addition, the surface of original PLA suture was very smooth, which led to lower stitching resistance when we were suturing the wound. Figure 4b shows that the coating of PCL/PGA wrapped tightly around the fibers, and then the surface of the PLA suture, coating with PCL/PGA, became very rough, which led to increased stitching resistance when we were suturing the wound. Moreover, the fibers in the suture were filled with the coating materials and glued together, which indicated that the coating of PCL/PGA improved the bundling ability of fibers in the suture.

In the process of degradation, the surface morphology of the original PLA suture and the PLA suture coating with 50/50 of PCL/PGA was changed, as shown in Figure 5.

Figure 5a–a” shows that with the extension of degradation process, the surface of the original PLA suture became more and more rough, which indicated that the original PLA suture gradually degraded from the surface of fiber. Figure 5b–b” shows that the coating material (PCL/PGA) of the suture was degraded firstly. When degraded for 13 weeks, a large number of holes, as shown in Figure 5b’, appeared on the coating. When degraded for 25 weeks, as shown in Figure 5b”, the coating material on the surface of suture was almost completely degraded, and then the internal PLA fibers began to degrade. In short, in the process of degradation, the suture gradually degraded from the coating materials to the inside fibers, and when degraded for about 25 weeks, there was only a small amount of materials degraded on the surface of original PLA suture, but the coating material of PLA suture coating with 50/50 PCL/PGA basically degraded completely.

The coating on the surface of PLA suture can be composed of different proportions of PCL/PGA, such as 70/30 and 30/70. The surface morphologies of PLA suture with different proportions of PCL/PGA were measured during degradation process, as shown in Figure 6.

When the PLA sutures were degraded for 13 weeks (Figure 6a,b), there was more coating materials on the suture with 70/30 of PCL/PGA than that on the suture with 30/70 of PCL/PGA. When degraded for 25 weeks (Figure 6a’,b’), there was many coating materials on the suture with 70/30 of PCL/PGA, but the coating materials on the suture, coating with 30/70 of PCL/PGA, was almost completely degraded, and the internal PLA fibers began to degrade. The above phenomena indicate that the surface coating of the suture, coating with 30/70 of PCL/PGA, degraded more quickly than that of the suture, coating with 70/30 of PCL/PGA. This further indicates that the more PCL in the coating, the slower degradation rate of the coating. This is because that the degradation rate of PCL is lower than that of PGA.

### 3.2. Chemical Structure of Drug-Loaded PLA Suture

The infrared spectra of sutures, coating with different proportions of PCL/PGA, are shown in Figure 7.

Figure 7a shows that the suture, coating with pure PCL, had many characteristic peaks, such as the asymmetric expansion vibration peak of –C–O–C at 1168 cm^−1^, the expansion vibration peak of –C=O at 1730 cm^−1^ and the expansion vibration peak of –CH at 2956 cm^−1^. Figure 7g shows the suture, coating with pure PGA, had many characteristic peaks, such as the bending vibration peak of –CH_2_ at 1074 cm^−1^ and the stretching vibration peak of –C=O at 1752 cm^−1^. The sutures, coating with mixed PCL/PGA in 90/10, 70/30, 50/50, 30/70 and 10/90 as shown in Figure 7b–f, contained the characteristic peaks of PGA and PCL, which did not shift, and did not appear new peaks obviously. This indicated that these materials, such as the PLA suture, the PCL/PGA carriers and the CPFX drug, were just physically mixed rather than chemically reacted. This did not change the chemical structure of the CPFX drug, which was very useful for ensuring the original efficacy of CPFX drug.

### 3.3. Mechanical Properties of Drug-Loaded PLA Suture

The drug-PCL/PGA was coated on the surface of PLA suture, and the proportion of PCL/PGA was an important factor which affected the mechanical properties of suture, as shown in Figure 8.

Figure 8 shows that with the increasing of PCL in the carriers, the breaking strength and the breaking elongation of suture both increased. This is because the mechanical property of pure PCL was more excellent than that of pure PGA, therefore, with the increase of PCL in the carriers, both the breaking strength and the breaking elongation gradually improved.

During the degradation process of sutures, the breaking strength of sutures, coating with different proportions of PCL/PGA, was changed, as shown in Figure 9.

Figure 9 shows that at 0 weeks, the breaking strength of original PLA suture (h) was larger than that of these sutures coated with PCL/PGA (a–g), which is because that the original PLA suture was damaged by the solvent erosion or the mechanical action during the pretreatment and dip-padding process, and then the breaking strength of sutures reduced.

It is also shown from Figure 9 that in the process of degradation with the prolonging of time, the breaking strength of all sutures (a–h) gradually decreased, and the strength reduction rate of original PLA was faster than that of the sutures coated with drug-PCL/PGA, which indicated that the degradation rate of original PLA suture (h) was much faster than that of other sutures (a–g). This is because with the gradual degradation of sutures, the structure of sutures was destroyed, and then their breaking strength decreased inevitably. Moreover, the suture gradually degraded from the coating materials to the inside fibers, and the coating materials of drug-PCL/PGA took some time to degrade, which caused that the structure of original PLA suture without any coating materials would damage more severely than that of these sutures coating with drug-PCL/PGA. Therefore, the degradation rate of original PLA suture was much faster than that of other sutures. Figure 5 also provides evidences to support this perspective.

In addition, when the breaking strength of sutures falls to a certain extent, the suture was broken under the wound tissue tension, and then lost its effect of tightening the wound. Therefore, in order to maintain the effectiveness of tightening wound, the strength of suture should be stronger than the certain standard of strength. For instance, the strength of skin injury suture is generally required more than 600 cN, so the 600 cN is the standard of strength (red dotted line in Figure 9). The corresponding time of intersections between red dotted line and other strength lines are the effective strength-time of tightening wound for the suture, as shown in Table 1, which indicated that with the increase of PCL in the coating, the effective strength-time increased.

### 3.4. Drug-Release Behavior of Drug-Loaded PLA Suture

The drug carrier of PCL/PGA, as the main coating material for the suture, was degraded gradually in the process of suture degradation. Meanwhile, the drug of ciprofloxacin (CPFX), which was carried by the PCL/PGA, was released from the carrier. The accumulated rate of drug-release was applied to evaluate the drug-release behavior of drug-loaded PLA suture, as shown in Figure 10.

Figure 10 shows that the drug-release rate of all sutures was very fast at the initial stage of degradation (within 100 h), and then after 100 h, the drug-release rate of sutures slowed down. Because the suture contained a lot of CPFX drug in the early stage of releasing, the drug of CPFX was very easy to release from the PCL/PGA carrier. As a result, the drug-release rate of sutures was very fast in the early stage (within 100 h). In the later stage (after 100 h), with the slow degradation of PCL/PGA carrier, the drug was released slowly from the drug-carrier of PCL/PGA.

Figure 10 also shows that the more PCL in the drug-carrier, the smaller accumulated rate of drug-release, which means a lower rate of drug releasing for the suture. This is because that the degradation rate of PCL was lower than that of PGA. Therefore, with the increasing of PCL in the drug-carrier, both the degradation rate and the drug-release rate reduced.

Different drug-release models were used to characterize the release mechanism of drugs [24,25,26]. The drug-release behavior was fitted by several classical drug release models, such as zero order model: *Q*_t_ = *Kt*, first-order drug release model: ln (1 − *Q*_t_) = −*Kt*, Higuchi model: *Q_t_* = *Kt*^1/2^, Ritger−Peppas model: *Q_t_* = *Kt*^n^, Hixon−Crowell model: (1 − *Q_t_*) ^1/3^= *Kt*, Weibull distribution curve [27,28,29]. The data obtained from the release of ciprofloxacin in the buffer solution were simulated in this experiment. It is found that the release curve of CPFX in Figure 10 conforms to the Ritger−Peppas model of *Q_t_* = *Kt*^n^, as shown in Table 2.

According to some literatures [30,31], the mechanism of drug-release was drug diffusion, when *n* ≤ 0.45. When 0.45 < *n* < 0.89, the drug diffusion and the dissolution of drug carrier have a synergistic effect on the release rate of drug. The drug release fits with the skeleton dissolution mechanism when *n* ≥ 0.89. It can be seen from Table 2 that the value of “*n*” is between 0.45 and 0.89, so the drug release mechanism of CPFX-PCL/PGA, in this paper, is a synergistic effect of drug diffusion and PCL/PGA carrier dissolution.

## 4. Conclusions

The fibers in original PLA suture were very loose, and its surface was very smooth. The surface of the PLA suture, coating with PCL/PGA, was very rough, which led to increased stitching resistance. In the process of degradation, the suture gradually degraded from the coating materials to the inside fibers. With the increasing of PCL in the coating of suture, the degradation rate of the coating goes down. The infrared spectra of sutures shows that these coating materials were just physically mixed and did not form any new chemical bonds, which was very useful for ensuring the original efficacy of CPFX drug. With the increasing of PCL in the carriers, both the breaking strength and the breaking elongation of suture increased. The breaking strength of original PLA suture was larger than that of these sutures coated with PCL/PGA. During degradation, the breaking strength of all sutures gradually decreased, and the more PCL in the coating materials, the longer effective strength-time for the suture. The drug-release rate of all sutures was very fast at the initial stage of degradation (within 100 h), and then the drug-release rate of sutures began to slow down (after 100 h). With the increasing of PCL in the drug-carrier, the rate of drug releasing became lower. The drug release mechanism of CPFX-PCL/PGA, in this paper, was a synergistic effect of drug diffusion and PCL/PGA carrier dissolution.

## Figures and Tables

**Figure 1 polymers-12-00288-f001:**
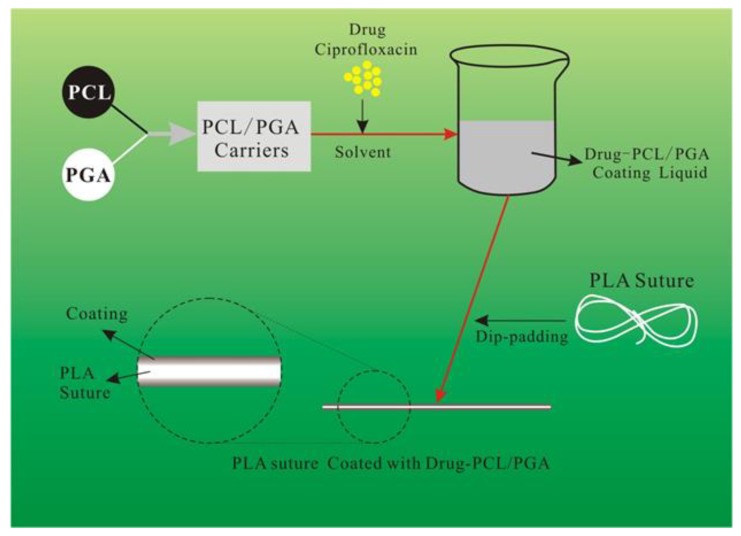
The whole scheme in this paper.

**Figure 2 polymers-12-00288-f002:**
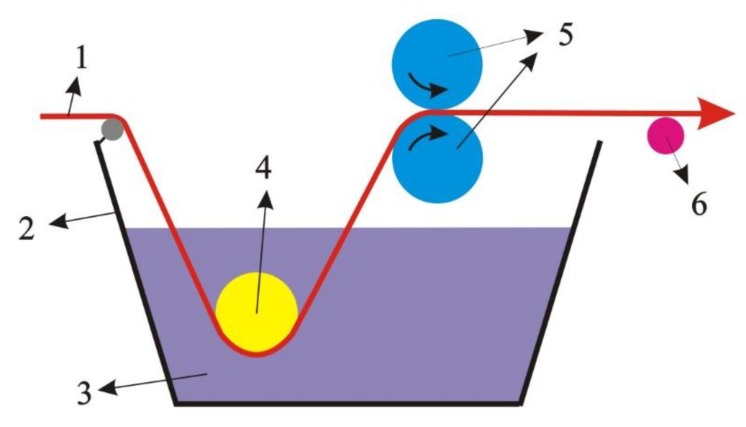
Process of dip-padding. (**1**) Pretreated polylactic acid (PLA) surgical suture; (**2**) dip-padding tank; (**3**) coating finishing liquid of ciprofloxacin (CPFX)-polycaprolactone (PCL)/polyglycolide (PGA); (**4**) dipping rollers; (**5**) rubber press rollers; (**6**) guide roller.

**Figure 3 polymers-12-00288-f003:**
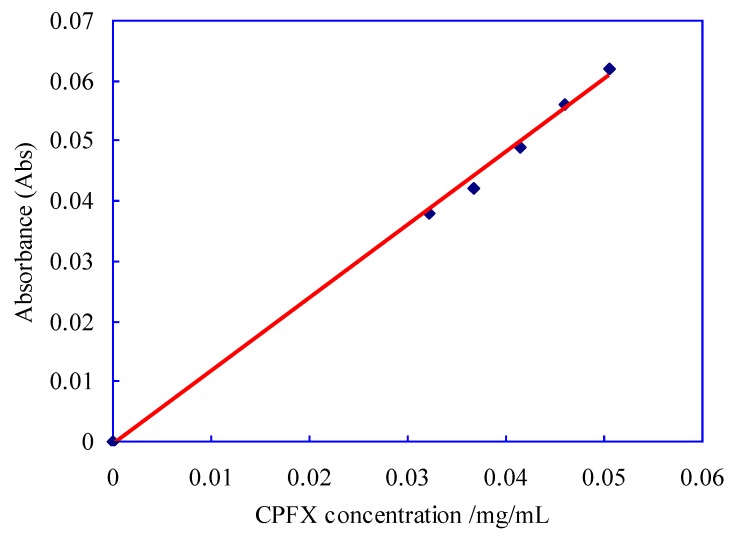
Standard curve of ciprofloxacin (CPFX) drug.

**Figure 4 polymers-12-00288-f004:**
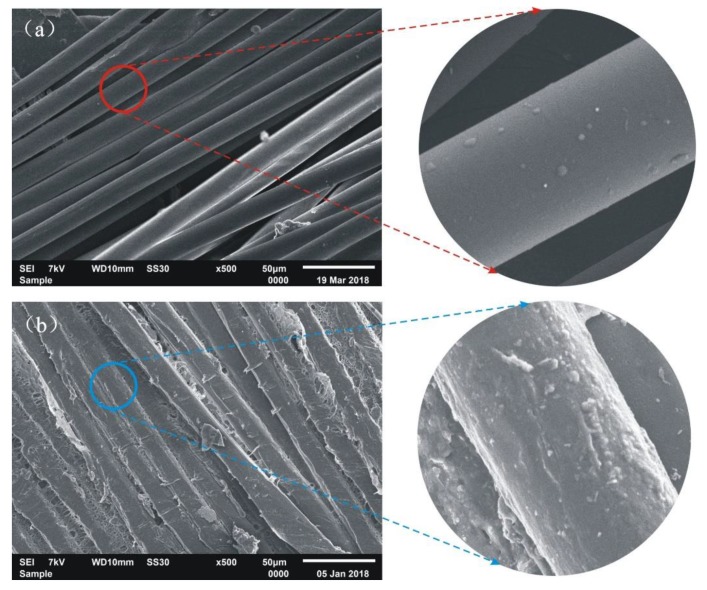
SEM images of suture. (**a**) Original PLA suture; (**b**) PLA suture coating with 50/50 of PCL/PGA.

**Figure 5 polymers-12-00288-f005:**
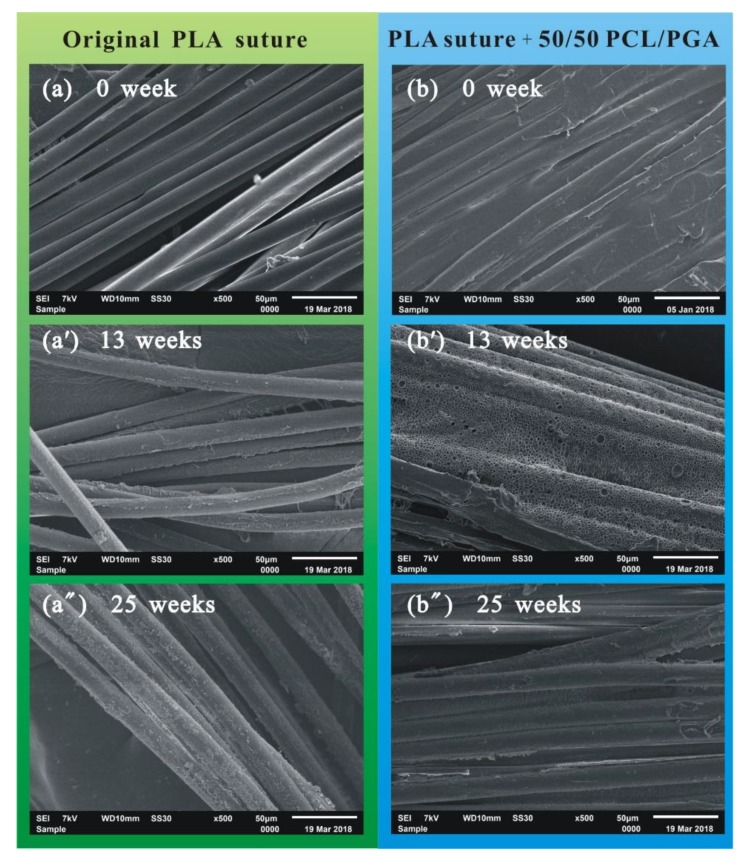
SEM images of sutures. (**a**) Original PLA suture degraded for 0 week; (**a’**) original PLA suture degraded for 13 weeks; (**a”**) Original PLA suture degraded for 25 weeks; (**b**) PLA suture coating with 50/50 of PCL/PGA degraded for 0 weeks; (**b’**) PLA suture coating with 50/50 of PCL/PGA degraded for 13 weeks; (**b”**) PLA suture coating with 50/50 of PCL/PGA degraded for 25 weeks.

**Figure 6 polymers-12-00288-f006:**
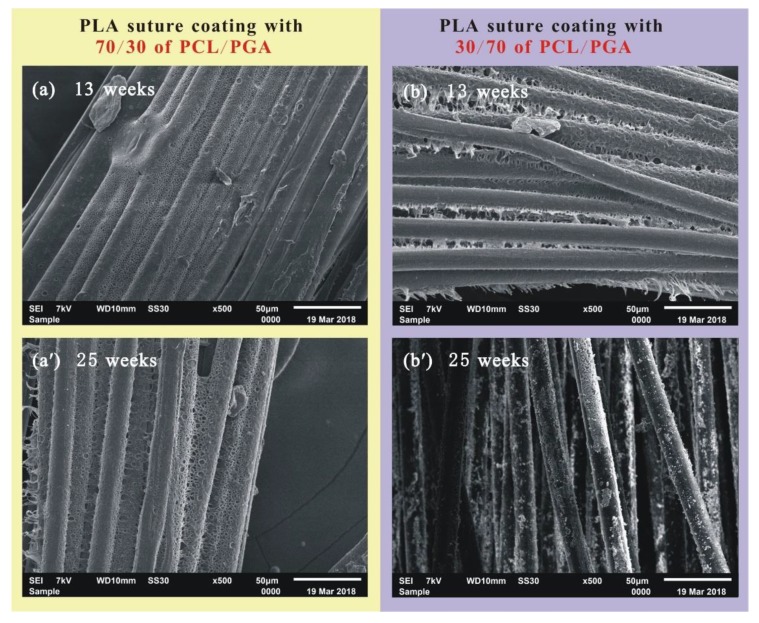
SEM images of suture. (**a**) PLA suture coating with 70/30 of PCL/PGA degraded for 13 weeks; (**a’**) PLA suture coating with 70/30 of PCL/PGA degraded for 25 weeks; (**b**) PLA suture coating with 30/70 of PCL/PGA degraded for 13 weeks; (**b’**) PLA suture coating with 30/70 of PCL/PGA degraded for 25 weeks.

**Figure 7 polymers-12-00288-f007:**
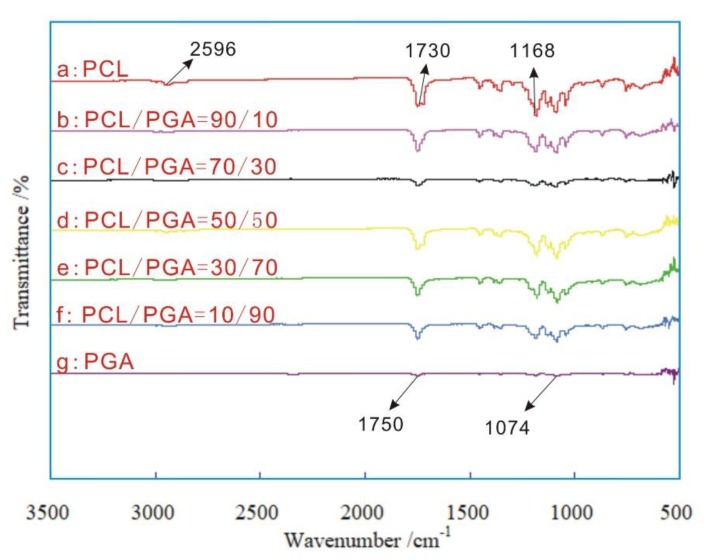
Infrared spectra of sutures coating with different proportions of PCL/PGA.

**Figure 8 polymers-12-00288-f008:**
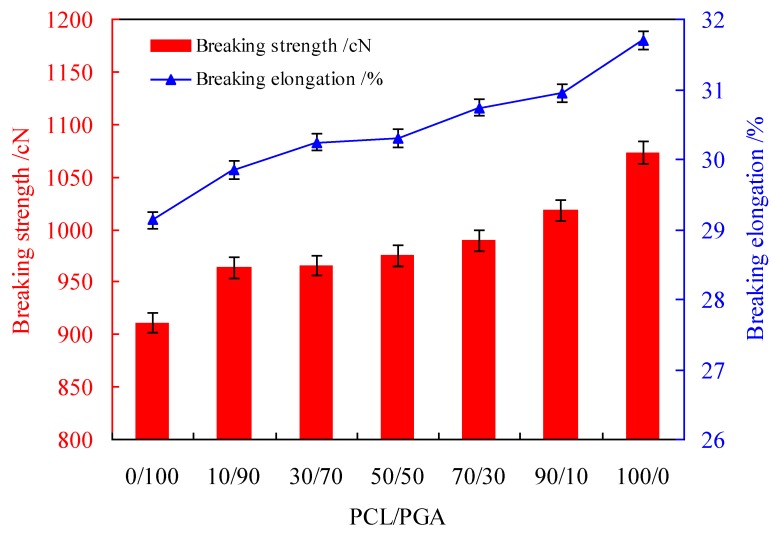
The breaking strength and elongation of the sutures coating with different proportions of PCL/PGA.

**Figure 9 polymers-12-00288-f009:**
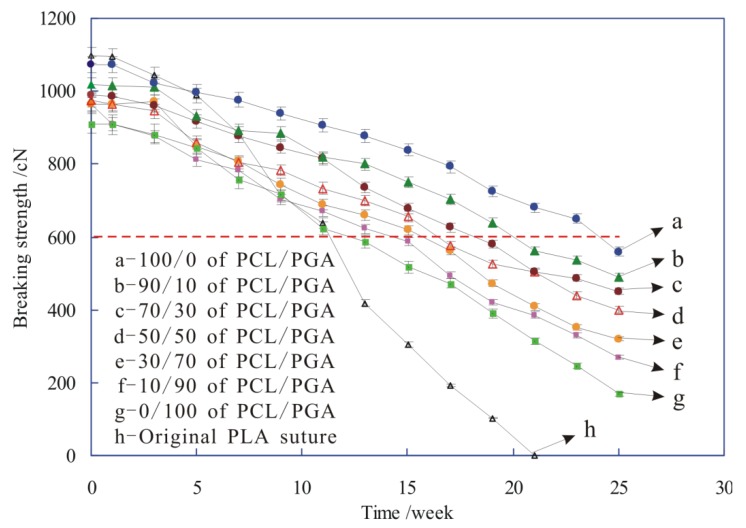
The breaking strength of sutures during degradation.

**Figure 10 polymers-12-00288-f010:**
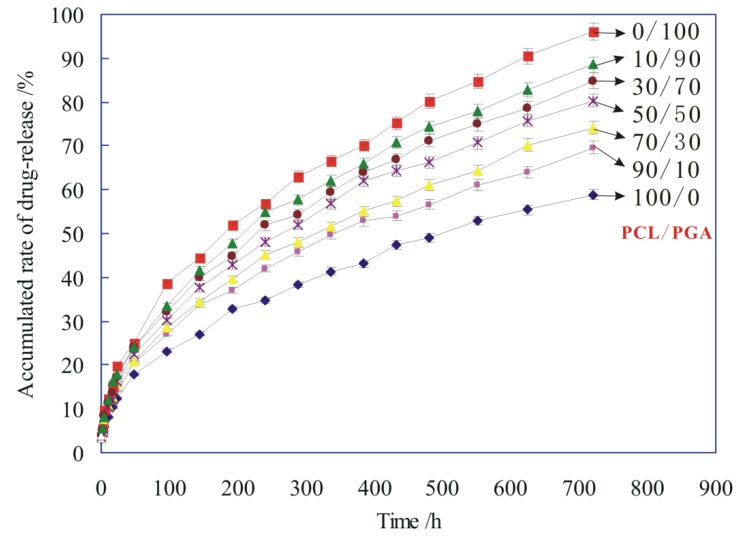
The accumulated rate of drug-release of sutures with different proportions of PCL/PGA carriers.

**Table 1 polymers-12-00288-t001:** The effective strength-time of tightening wound for the sutures.

PCL/PGA	Original PLA Suture	0/100	10/90	30/70	50/50	70/30	90/10	100/0
Effective strength-time /weeks	11.3	12.6	14.5	15.8	16.6	18.4	20.1	24.2

**Table 2 polymers-12-00288-t002:** Fitting parameters of drug release Ritger−Peppas model of *Q_t_* = *Kt*^n^.

PCL/ PGA		100/0	90/10	30/70	50/50	30/70	10/90	0/100
**Ritger−Peppas Equation:** $Q_{t}=Kt^{n}$	*K*	2.7829	3.3780	3.4313	4.1898	3.4539	3.1845	4.0301
	*n*	0.4644	0.4587	0.4665	0.4765	0.4797	0.4718	0.4703
	*R* ^2^	0.9992	0.9993	0.9994	0.9994	0.9993	0.9996	0.9994

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
