# Peer review of "Controllable Drug Release Behavior of Polylactic Acid (PLA) Surgical Suture Coating with Ciprofloxacin (CPFX)—Polycaprolactone (PCL)/Polyglycolide (PGA)"

_polymers, 2020, doi:10.3390/polym12020288_

Round 1

Reviewer 1 Report

The manuscript entitled ”Controllable drug release behavior of polylactic acid  (PLA) surgical suture coating with ciprofloxacin  (CPFX) - polycaprolactone (PCL) / polyglycolide (PGA)” by Liu et al. described the release rate of drug (CPFX) can be controlled by changing the ratio of PCL/PGA. It be found that the drug release mechanism of CPFX-PCL/PGA was a synergistic effect of drug diffusion and PCL/PGA carrier dissolution. There are several significant issues that have to address in order to be acceptable for publication. 

Comments:

1. Line 113, page 3: Please indicate the purity of alcohol and the recipe of medical alcohol. 2. Page 4: In the section 2.3, the coated sutures were placed in a jar which was containing 50 mL PBS buffer solution at 37oC. The experimental time was about 25 weeks. Do they need to add antibiotics avoid bacteria grow in PBS buffer? Please explain more details in the experiment procedure.3. Page 9: In Fig. 8, the error bars are same each data. Please explain how many samples were tested each data. 4. Please also fill the error bars of both Figs. 9 & 10.

Author Response

Dear Reviewer:

Thank you for your comments on our manuscript entitled " Controllable drug release behavior of polylactic acid (PLA) surgical suture coating with ciprofloxacin (CPFX) - polycaprolactone (PCL) / polyglycolide (PGA) (ID: 683196)”.

Those comments are very helpful for revising and improving our paper. We have studied the comments carefully and made corresponding revisions which we hope to meet with approval.

The main revisions in the manuscript and the responds to the reviewers’ comments are as follows:

Responses to the reviewer’ comments:

Reviewer 1

Comments and Suggestions for Authors:

Line 113, page 3: Please indicate the purity of alcohol and the recipe of medical alcohol.

Responds: Thanks for reviewer’s sensible advice. We had indicated in “2.1. Materials” part that the purity of absolute ethyl alcohol is 99.5%, and the purity of medical alcohol is 75%. And the modifications are marked in red in the revised paper.

Page 4: In the section 2.3, the coated sutures were placed in a jar which was containing 50 mL PBS buffer solution at 37oC. The experimental time was about 25 weeks. Do they need to add antibiotics avoid bacteria grow in PBS buffer? Please explain more details in the experiment procedure.

Responds: We are grateful for reviewer’s helpful suggestion. We didn’t add antibiotic in PBS buffer, but the experiment was conducted in a sterile room, where 12 hours of UV light per day was used to kill bacteria, to prevent the growth of bacteria. We added a more detailed description of the experiment in the paper, and the modifications are marked in red.

Page 9: In Fig. 8, the error bars are same each data. Please explain how many samples were tested each data.

Responds: In Fig.8, 20 samples were tested each data. The error bars are not the same, however, only because the error gap is relatively small, the gap among error bars is not obvious at first glance. Thank you very much.

Please also fill the error bars of both Figs. 9 & 10.

Responds: According to the suggestion of reviewer, we had filled the error bars in Fig.9 and Fig.10. Thank you very much for helpful suggestion.

Reviewer 2 Report

The research topic is interesting, properly presented. My suggestion is to do antimicrobial testing of the material.

Author Response

Dear Reviewer:

Thank you for your comments on our manuscript entitled " Controllable drug release behavior of polylactic acid (PLA) surgical suture coating with ciprofloxacin (CPFX) - polycaprolactone (PCL) / polyglycolide (PGA) (ID: 683196)”.

Those comments are very helpful for revising and improving our paper. We have studied the comments carefully and made corresponding revisions which we hope to meet with approval.

The main revisions in the manuscript and the responds to the reviewer’ comments are as follows:

Responses to the reviewer’ comments:

Reviewer 2

Comments and Suggestions for Authors:

The research topic is interesting, properly presented. My suggestion is to do antimicrobial testing of the material.

Responds: The main purpose of this paper is to control the drug release rate of suture, rather than the antibacterial property. The Ciprofloxacin (CPFX) is used only as a proxy for the drug, and its release performance from suture is the research focus of this paper. Therefore, this paper focuses on the drug release behavior, rather than the antibacterial property. Of course, we will continue to study the antibacterial property of drugs in the following work. Thanks for your suggestions.

Reviewer 3 Report

The review under consideration concerns study of drug release behavior of polylactic acid surgical suture coating with ciprofloxacin - polycaprolactone / polyglycolide.

The choosing of the topic are very interesting, manuscript is well written but needs corrections:

Please check English throughout the manuscript. (line 38 – biodegradable, line 87- ware, line 175 future tense …etc.) Line 94: Here You should use only abbreviation. Please check and correct in the manuscript. Line 96: check the Mw value of PGA Line 118: Remove the (CH3COOC2H5) add the purity of ethyl acetate Line 120: drug-CPFX this way of writing suggests some connection, please change it (e.g. drug or drug (CPFX)). Line 142: SEM is not a property. Line 155-156: Is this a normal procedure? Do You have some reference to that?

After corrections the presented research may be suitable for publication in Polymers.

Author Response

Dear Reviewer:

Thank you for your comments on our manuscript entitled " Controllable drug release behavior of polylactic acid (PLA) surgical suture coating with ciprofloxacin (CPFX) - polycaprolactone (PCL) / polyglycolide (PGA) (ID: 683196)”.

Those comments are very helpful for revising and improving our paper. We have studied the comments carefully and made corresponding revisions which we hope to meet with approval.

The main revisions in the manuscript and the responds to the reviewers’ comments are as follows:

Responses to the reviewer’ comments:

Reviewer 3

Comments and Suggestions for Authors:

Please check English throughout the manuscript. (line 38 – biodegradable, line 87- ware, line 175 future tense …etc.) Line 94: Here You should use only abbreviation. Please check and correct in the manuscript. Line 96: check the Mw value of PGA Line 118: Remove the (CH3COOC2H5) add the purity of ethyl acetate. Line 120: drug-CPFX this way of writing suggests some connection, please change it (e.g. drug or drug (CPFX)). Line 142: SEM is not a property.

Responds: According to the suggestions of reviewer, we had checked and corrected the manuscript, and the modifications are marked in red in the revised paper. Thank you very much for your careful review.

Line 155-156: Is this a normal procedure? Do You have some reference to that?

Responds: Yes, this is a normal procedure in Line 155-156. We had added two references to that. We are grateful for your helpful suggestion.

Round 2

Reviewer 3 Report

Line 99 and 100:please check again the Mw value of PGA and PCL, the Mw value of PGA seems very high,  maybe it's a matter of units

Author Response

Dear Reviewer:

Thank you for your comments on our manuscript entitled " Controllable drug release behavior of polylactic acid (PLA) surgical suture coating with ciprofloxacin (CPFX) - polycaprolactone (PCL) / polyglycolide (PGA) (ID: 683196)”.

Those comments are very helpful for revising and improving our paper. We have studied the comments carefully and made corresponding revisions which we hope to meet with approval.

The main revisions in the manuscript and the responds to the reviewers’ comments are as follows:

Responses to the reviewer’ comments:

Reviewer 3

Comments and Suggestions for Authors:

Line 99 and 100:please check again the Mw value of PGA and PCL, the Mw value of PGA seems very high,  maybe it's a matter of units.

Responds: Thank you very much for your careful review. The units of Mw value of PGA and PCL were wrong, that is, the “kg/mol” should be “g/mol”. Finally, the Mw of PGA is 11602 g/mol and the Mw of PCL is 50000 g/mol. Thanks again for your helpful suggestion.

Round 3

Reviewer 3 Report

Accept in present form.